# Functional Exploitation of Carob, Oat Flour, and Whey Permeate as Substrates for a Novel Kefir-Like Fermented Beverage: An Optimized Formulation

**DOI:** 10.3390/foods10020294

**Published:** 2021-02-01

**Authors:** Sana M’hir, Pasquale Filannino, Asma Mejri, Ali Zein Alabiden Tlais, Raffaella Di Cagno, Lamia Ayed

**Affiliations:** 1Laboratory of Microbial Ecology and Technology (LETMi), National Institute of Applied Sciences and Technology (INSAT), University of Carthage, BP 676, 1080 Tunis, Tunisia; asmamejri8@gmail.com (A.M.); lamiaayed@yahoo.fr (L.A.); 2Department of Animal Biotechnology, Higher Institute of Biotechnology of Beja, University of Jendouba, BP 382, 9000 Beja, Tunisia; 3Department of Soil, Plant and Food Science, University of Bari Aldo Moro, 70126 Bari, Italy; pasquale.filannino1@uniba.it; 4Faculty of Sciences and Technology, Libera Università di Bolzano, 39100 Bolzano, Italy; AliZeinAlabiden.Tlais@natec.unibz.it (A.Z.A.T.); raffaella.dicagno@unibz.it (R.D.C.)

**Keywords:** carob, oat flour, whey permeates, response surface method, kefir grain, fermentation, lactic acid bacteria, phenolics

## Abstract

This study investigated the fortification of a carob-based kefir-like beverage (KLB) with whey permeate (WP) and oat flour (OF). The response surface method was used to show the effect of WP and OF concentrations on lactic acid bacteria and yeast cell densities, pH, total titratable acidity (TTA), total phenolics content (TCP), DPPH radical scavenging activity, and overall acceptability (OA) in KLB. The statistical design provided thirteen formulations where OF concentration varied from 3% to 5% and WP from 10% to 15%. The enrichment of carob pods decoction with WP and OF had a positive effect on biomass production. Overall fermentation was shown to increase TPC of KLB. Furthermore, OF supplementation led to the higher levels of TPC and antiradical activity. WP negatively affected OA at linear and quadratic levels, whereas no effect of OF was observed at the linear level. The optimum point was found by using WP at 11.51% and OF at 4.77%. Optimized KLB resulted in an enrichment of bioavailable phenolics derivatives and highly digestible proteins.

## 1. Introduction

Functional beverages represent one of the fastest-growing segments in a novel food category and an attractive topic of study for researchers [1]. Milk kefir is an ancient fermented beverage, and it has been associated to the framework of functional foods due to its recognized health-promoting activities, such as antitumoral, cholesterol-lowering, and immunomodulatory effects [2]. Several bacteria and yeasts isolated from kefir were also classified as potential probiotics [3,4,5]. Ingestion of probiotics may exert a positive repercussion on human health, through several mechanisms interfering with the absorption of nutrients, the immune response, and the maintenance of the intestinal barrier functions [6]. Kefir grains consist of a polysaccharide matrix, which vehicles a complex microbial consortium mainly composed of lactic acid bacteria in symbiotic association with yeasts and acetic acid bacteria [7]. They ferment sugar substrates to produce organic acids, CO_2_, ethanol, and multiple volatile flavor [8]. Different biotechnological innovations were previously proposed to develop a kefir-like beverage with improved sensory, nutritional, and functional features. For instance, kefir grains were used to ferment non-dairy substrates, such as tea and plant juices, to produce functional beverages enriched in bioactive compounds or able to vehicle potential health-promoting bacteria [9]. Under the conditions of our study, we investigated the fortification of a carob-based kefir-like beverage (KLB) with WP and OF. Carob fruit is known to prevent some chronic diseases and to provide human health benefits (e.g., cholesterol-lowering, anticancer, hypoglycemic, and antimicrobial effects) [10,11]. It is a rich source of natural bioactive compounds such as dietary fiber and polyphenols [12]. Its antioxidant capacity is mainly related to its high level of phenolic compounds, including flavonoids and condensed tannins [13]. The high content of fermentable sugars make carob a suitable substrate for the growth of lactic bacteria and yeasts, although the deficiency in proteins and fatty acids might make nutritional supplements necessary for microbial growth [14,15,16]. Fortification of functional beverages with prebiotic such as OF [17] might support the growth of microorganisms during fermentation other than by providing nutrients and bioactive compounds for humans. Oat is a valuable source of unsaturated fatty acids [18] and low-cost proteins [19]. Epidemiological studies linked the consumption of oat grains to a lower risk of diabetes, obesity, and certain cancers [20]. These beneficial effects would be related to the synergy between fibers, antioxidants, vitamins, and minerals. Finally, the positive contribution of lactic acid bacteria on nutritional and sensory properties of cereal-based matrices due to the increased phenolics availability, antioxidant activity, and overall intensity of odor and flavor should be not overlooked [21]. WP, a by-product of the cheese-making industry, was previously described as valuable growth medium for probiotic bacteria, able to ensure a high microbial viability [22]. The use of WP as natural antioxidant in foods was also proposed [23]. Furthermore, the addition of whey to fermented products allows the enhancement of prebiotic properties [24]. 

Since the functional potential of KLB results from the interactions between different matrices and microbial activities, we used the response surface methodology (RSM) to optimize the KLB formulation.

The aim of this study was to optimize the formulation of a functional KLB, which was fortified with various amount of WP and OF to improve the health-promoting features and sensory properties of KLB. A central composite design (CCD) was used to analyze the interactions between the independent variables, producing a mathematical model describing the investigated system. This approach allowed the determination of the optimal WP and OF supplementation levels to maximize the cell density of lactic acid bacteria and yeasts, the DPPH radical scavenging activities, the total phenolics content, and the consumer acceptability of KLB. Phenolic profile and protein digestibility were further evaluated for the optimized beverage.

## 2. Material and Methods

### 2.1. Raw Materials and KEFIR Grains

Kefir grains belonged to LETMi laboratory (Tunis, Tunisia) and consisted of a symbiotic consortium of lactic acid bacteria (*Leuconostoc* spp., *Lactobacillus* spp., and *Lactococcus* spp.) and yeasts (*Saccharomyces* spp. and *Zygosaccharomyces* spp.), as previously characterized [25]. Grains were routinely propagated in pasteurized cow’s milk at room temperature (ca. 25 °C). After filtration, grains were collected and used to inoculate the mixtures [26]. 

Whey permeates (WP) was supplied by a dairy industry (Béja, Tunisia) and used at different concentration ranging from 10% to 15% (*w*/*v*) (WP composition: proteins 3%, lactose 85%, and ash 7%). Oat flour (OF) was obtained from a local supermarket (Tunis, Tunisia) (14.8% proteins, carbohydrates 66.3%) and used at different concentrations ranging from 3% to 5% (*w*/*v*). Carob decoction (15 °Brix) was obtained by boiling carob pods (100 g carob pods/150 mL distilled water) for 30 min. After cooling, the decoction was filtered with Whatman paper and stored until uses.

### 2.2. KLB Preparation

KLB formulation (Table 1 and Table 2) was developed using carob decoction (50%) and a mixture of WP and OF (50%). The WP–OF suspension was prepared by adding the corresponding amounts of WP and OF to distilled water. All beverages were pasteurized at 90 °C for 5 min before the addition of kefir grains. An overnight grown culture of kefir grain was inoculated (5%, *v*/*v*) into 300 mL of KLB. The suspensions were fermented in 500 mL Erlenmeyer flasks at 30 °C. Samples were drawn at 24 h and analyzed for viability of lactic acid bacteria and yeasts, pH, total titratable acidity (TTA), total phenolics content (TPC), and DPPH radical scavenging activity.

### 2.3. Experimental Design

Whey permeates and oat flour concentration were the two variables used to study the interaction and the quadratic effects on pH, TTA, lactic acid bacteria and yeast growth, TPC, DPPH radical scavenging, and the overall acceptability (OA). Thirteen experiments were supported by using a central composite design (CCD) at five coded levels of −1.41, −1, 0, 1, and 1.41 (Table 1 and Table 2) with five-repeated central point [27]. The responses’ functions (Y) were related to the coded variables (xi, i = 1, 2) by a second-order polynomial using this equation: Y = b_0_ + b_1_ X_1_+b_2_ X_2_ +b_12_ X_12_ + b_11_ X_1_^2^ + b_22_ X_2_^2^(1)

Coefficients were b_0_ (constant), b_1_, b_2_ (linear), b_11_, b_22_ (quadratic), and interaction b_12_ of the model and were calculated by NEMROD-W software (version 99901, LPRAI Company).

The effect of independent variables was explained according the following equations:Lactic acid bacteria = 9.180 + 0.171 X_1_ + 0.161 X_2_ −0.466 X^2^_1_ − 0.473 X^2^_2_; *R^2^* = 0.986(2)
Yeasts = 6.051 + 0.131 X_1_ + 0.080 X_2_ −0.175 X^2^_1_ − 0.146 X^2^_2_ –0.043 X_1_X_2_; *R^2^* = 0.925(3)
pH = 4.258 + 0.047 X_1_ + 0.032 X_2_ +0.035 X^2^_1_ +0.012 X^2^_2_; *R^2^* = 0.861(4)
TTA = 1.054 + 0.029 X_1_ − 0.078 X^2^_1_ − 0.08 X^2^_2_ − 0.032 X_1_X_2_; *R^2^* = 0.920 (5)
TPC = 14.556 + 3.866 X_2_ +1.893 X^2^_1_ + 3.624 X^2^_2_ + 1.094 X_1_X_2_; *R^2^* = 0.835(6)
DPPH= 73.211 − 0.855 X_1_ + 4.167 X_2_ + 2.255 X^2^_1_ + 1.072 X^2^_2_ + 2.557 X_1_X_2_; *R^2^* = 0.816(7)
OA= 3.464 − 0.163 X_1_ − 0.070 X^2^_1_ − 0.170 X^2^_2_ − 0.20 X_1_X_2_; *R^2^* = 0.965(8)

### 2.4. Determination of pH and TTA

The values of pH were determined by a pH-meter (Mettler-Toledo EL20) with a food penetration probe. TTA was determined on 10 g of kefir homogenized with 90 mL of distilled water and using 0.1 N NaOH to reach pH 8.1 and expressed as percentage (*w*/*w*) of lactic acid [28].

### 2.5. Microbiological Analyses

Lactic acid bacteria were enumerated on MRS (de Man, Rogosa, and Sharpe) agar plates containing cycloheximide (0.005%, *w*/*v*) using serial dilution. Plates were incubated at 37 °C for 48 h. Yeasts were counted on potato dextrose agar (PDA) with chloramphenicol (500 µg/mL) to inhibit the growth of bacteria. Viable counts were enumerated after incubation at 30 °C for 72 h. Results were expressed as log of colony-forming units per ml of fermented beverage (log CFU/mL).

### 2.6. Determination of Total Phenolic Compounds (TPC)

The method of the Folin–Ciocalteau was used by mixing 100 µL of diluted KLB with 500 µL reagent. After shaking, samples were incubated for 3 min and then 1.250 mL of sodium carbonate (20%, *w*/*v*) were added. The mixtures were incubated for 60 min at room temperature. The absorbance was measured at 750 nm. Results were expressed as mg gallic acid equivalents per mL of beverage [29].

### 2.7. DPPH Radical Scavenging Activity 

Radical scavenging activity was assayed using the radical 2,2-diphenyl-1-picrylhydrazyl (DPPH) as reported by [30]. Seven-hundred µL of methanolic solution of DPPH were added to 700 µL of sample. After mixing and keeping at room temperature for 30 min, the absorbance was measured at 517 nm. The antioxidant activity was recorded as DPPH radical scavenging activity (%).

### 2.8. Consumer Rating of Sensory Attributes

A group of 60 people (33 male, 27 female, and age 20–49) were requested to provide their score for color, taste, odor (aroma intensity), texture (consistency), and overall acceptability (OA) according to an increasing hedonic scale from 1 (extremely disliked) to 5 (extremely liked). An aliquot of 20 mL of KLB was shown in transparent bottles. 

The average score for each KLB formulation was computed and used as a response for optimization.

### 2.9. Analyses of Phenolics Profiles 

The phenolics profile of KLB was investigated through liquid chromatography electrospray ionization tandem mass spectrometry (LC-ESI-MS/MS) analysis, according to the method previously validated by Tlais et al. [31]. A UHPLC Dionex 3000 (Thermo Fisher Scientific, Germany), was used, coupled to a TSQ Quantum™ Access MAX Triple Quadrupole Mass Spectrometer (Thermo Fisher Scientific, Germany), equipped with an electrospray source. Briefly, separation was with a Waters Acquity HSS T3 column (1.8 μm, 100 mm × 2.1 mm) (Milford, MA, USA), kept at 40 °C. Mobile phase A was water containing 0.1% formic acid; mobile phase B was acetonitrile containing 0.1% formic acid. The flow was 0.4 mL min^−1^, and the gradient profile was: 0 min, 2% B; from 0 to 3 min, linear gradient to 20% B; from 3 to 4.3 min, isocratic 20% B; from 4.3 to 9 min, linear gradient to 45% B; from 9 to 11 min, linear gradient to 100% B; from 11 to 13 min, wash at 100% B; from 13.01 to 15 min, back to the initial conditions of 5% B. The injection volume was 3 μL. Target phenols were detectable under multiple reaction monitoring mode, and the compounds were identified based on their reference standard, retention time, and qualifier and quantifier ion (Appendix A).

Separation, identification, and quantification of microbial derivatives of phenolic acids were performed according to the method validated by Filannino et al. [32]. The HPLC system Ultimate 3000 (Dionex, Germering, Germany) was used, equipped with a Kinetex C18 Phenomenex (150 × 4.6 mm with a particle size of 5 μm) column (Thermo Fisher Scientific), a photodiode array detector (PAD 3000), low-pressure pump Ultimate 3000, and an injector loop Rheodyne (Rheodyne, USA, volume 20 μL). Briefly, the injected volume was 10 μL, and the column oven was set at 35 °C. Mobile phase consisted of (A) water + 0.1% trifluoroacetic acid (TFA) and (B) acetonitrile + 0.1% TFA. A linear gradient program at a flow rate of 1.0 mL min^−1^ was used: 0.0–5.0 min from 5% to 10% (B), 5.0–25 min from 10% to 40% (B), 25–45 min from 40% to 90% (B), then 90% (B) for 5 min and 45–50 min from 90% to 5% (B). PAD analyses of phenolic acid derivatives were performed at 280, 310, and 320 nm wavelengths (Appendix A).

Chromeleon Software version 7 (Dionex, Germering, Germany) was used for instrument control, data acquisition, and data analysis.

### 2.10. In Vitro Protein Digestibility

The in vitro protein digestibility was determined on KLB both prior and after the fermentation by the method proposed by Akeson and Stahmann [33] with some modifications [34]. Samples were subjected to a sequential enzyme treatment mimicking the in vivo digestion in the gastro-intestinal tract, and protein digestibility was expressed as the percentage of the total protein, which was solubilized after enzyme hydrolysis.

### 2.11. Statistical Analysis

Analyses were carried out in triplicate on three biological replicates for each condition. Data were subjected to analysis of variance (ANOVA) test for multiple comparisons (one-way ANOVA followed by Tukey’s procedure at *p* < 0.05), using the statistical software Statistica 7.0 (StatSoft, Tulsa, OK, USA).

## 3. Results and Discussion

### 3.1. Response Surface Methodology (RSM) Implementation

Carob is an excellent source of phytochemicals, including phenolic compounds [35]. We applied the RSM to optimize the formulation of a carob-based KLB fortified with WP and OF, aiming at increasing the growth and the viability of lactic acid bacteria and yeasts, the DPPH radical scavenging activity, and the total phenolics content. The process variables investigated during fermentations were WP and OF concentrations. WP was used due to its high content of oligosaccharides [24,36], whereas OF was used as a source of amino acids, unsaturated fatty acids, and dietary fibers [37,38]. Such components have been shown to exert health-promoting effects in humans [37,38], as well as playing a positive role in promoting the microbial growth during KLB fermentation and microbial viability as source of a potential probiotic activity [17,24]. 

The responses to the effects of WP (X_1_) and OF (X_2_) (each one at five levels and in the combinations) in terms of lactic acid bacteria and yeast cell densities, pH, TTA, TPC, DPPH radical scavenging activity, and OA, were shown in Table 2. Based on the statistical analysis, the proposed model resulted adequately, possessing no significant lack of fit and with very satisfactory values of the *R*^2^ for the majority of the responses. The closer the value of *R*^2^ to unity, the better the empirical models fit the actual data (Appendix A).

### 3.2. Effect of Independent Variables on Lactic Acid Bacteria and Yeasts Growth

Response surface was used to illustrate the effect of WP and OF concentrations on the growth of lactic acid bacteria and yeasts in KLB. The response surfaces and contour plots are shown in Figure 1A,B.

Lactic acid bacteria and yeast cell densities varied from 7.85 ± 0.08 to 9.29 ± 0.09 and 5.47 ± 0.27 to 6.05 ± 0.29 log CFU/mL, respectively (Table 2). The enrichment of carob decoction with OF and WP led to a significant improvement of microbial growth. When carob was used as sole carbon and nitrogen source (data not shown), the lactic acid bacteria growth was limited. Bouhadi et al. [15] previously showed an increase of lactic acid bacteria growth and lactate production when the carob syrup was enriched with sweet cheese. The promoting effect of oat on the growth of probiotic bacteria and their use as a probiotic carrier was also described by several authors [39,40,41]. The results of LAB and yeast counts were in accordance with those shown by Sabokbar et al. [42], using whey and apple juice as growth media.

The effects of WP and OF on growth of yeasts and lactic acid bacteria at linear, quadratic, and interaction levels are shown in Table 3. The sign of the coefficients for these parameters at the linear level is positive, confirming their synergistic effects. However, negative interaction was found between WP and OF on yeast growth (b12, *p* < 0.001). Quadratic effects were also negative for lactic acid bacteria and yeasts responses.

The three-dimensional response surface was used to understand the interaction between the two factors and to visualize the combined effects of factors on lactic acid bacteria and yeasts growth (Figure 1A,B). The shapes of the contour plots can explain the interaction between the variables [43]. Elliptical contour plots indicated the interactions. However, circular ones indicated that the interactions between the variables are negligible. Therefore, the interactions between OF and WP were negligible on lactic acid bacteria and yeast growth. 

### 3.3. Effect of Independent Variables on pH and TTA

The pH and TTA values ranged from 4.25 ± 0.02 to 4.39 ± 0.02 and 0.79% ± 0.02% to 1.07% ± 0.04% (*w*/*w*), respectively. WP affected positively the pH and TTA responses for linear effect (Table 3). According to Ismaiel et al. [44], lactose is the most suitable carbon source for kefir-associated microorganisms, giving the highest values of biomass and lactic acid concentration. Lactic acid is needed to ensure proper flavor and prolonged shelf life of the kefir beverage. However, the pH values are not correlated with the TTA, likely due to the different buffering capacity of WP and OF. The pH and TTA were found to be in function of the linear and quadratic effects of WP and OF. The linear and quadratic effects were both positive for pH, which explained the observed behavior of the curve, as shown in Figure 2A. As shown by linear coefficients, WP and OF positively affected the pH and TTA. Nevertheless, the quadratic coefficients of WP and OF were negative for TTA (Figure 2B). On the other hand, the interactive effect between WP and OF was not significant for pH, but it was significant for TTA (Table 3). 

### 3.4. Effect of Independent Variables on TPC and Antiradical Activity

The increase in TPC was linked to the linear effect of OF (*p* < 0.001), to the quadratic effect of both WP and OF (*p* < 0.05), and to the interactive effect of WP and OF (*p* < 0.001) (Table 3). Phenolic compounds of carob mostly comprise hydroxybenzoic acids, flavonols, flavan-3-ols, and gallotannins [45]. Although the richness of KLB in phenolic compounds is also attributable to the carob decoction, OF was found to have a significant positive effect on TPC. Oat is rich in phenolic compounds, especially ferulic acid [46]. Furthermore, avenanthramides, which are hydroxycinnamoyl anthranilate alkaloids, are found exclusively in oats [20,47,48]. Positive interaction was found between WP and OF on TPC response, as showed by elliptical contour plots (Figure 3A).

Analysis of variance (ANOVA) was carried out to determine the significance of WP and OF and their quadratic and interactive levels on TPC (Table 3). 

Health benefits of phenolic compounds are typically related to their antioxidant activity, which is associated with their ability to donate hydrogen or electrons to free radicals, offering protection against oxidative processes and chronic disease [49,50]. Linear, quadratic, and interactive effect of OF and WP on radical scavenging activity of KLB are significant. Linear and quadratic effect of OF are positive (b2 > 0 and b22 > 0) (Table 3). These results indicate that OF supplementation enhanced the antiradical activity of KLB, likely due to the abundance of antioxidant compounds in oat, which include not only phenolics, but also tocopherols, tocotrienols, and sterols.

WP negatively affected the DPPH radical scavenging activity (b1 < 0, *p* < 0.05). Mariken et al. [51] found that the antioxidant capacity of a mixture of phenolic compounds, and protein is less than the sum of the antioxidant capacity of phenolic and protein separately. However, the interactive levels of the variables WP and OF were positive (b12 > 0, *p* < 0.01), as shown by contour plots (Figure 3B). DPPH results showed high antioxidant activities, as reported by Randazzo et al. 2016 [52] 

KLB fermentation affected both the TPC and antiradical activity (Table 4). In all fermented KLB samples, the TPC values were significantly (*p* < 0.05) higher with respect to the unfermented samples. 

These results can be explained through the enzymatic activities of microorganisms (e.g., β-glycosidase, esterase), which are responsible for release of non-extractable phenolics [53]. The released metabolites likely led to an increase of the antiradical activity of KLB [30,53,54,55,56,57,58]. The experiment n°4 showed the highest levels of DPPH radical scavenging activity and TPC. In fact, DPPH radical scavenging activity and TPC increased from 68.84% to 85.7% and from 12.55 to 27.69 mg GAE/mL, respectively. Fermentation might be also responsible for releasing peptides or amino acids from WP, which are able to donate hydrogen atoms to DPPH radicals [59].

### 3.5. Effect of Independent Variables on CONSUMER Rating of Sensory Attributes

The influence of OF and WP, alone or combined, on the OA of KLB samples is represented in Table 3. The *R^2^* of models (Table 3) was close to 1 (0.965), indicating that variation in responses could adequately be explained by the concentration of ingredients. Furthermore, the statistical analysis of variance showed a non-significant lack of fit for the models, suggesting that the terms included in the models are sufficient to describe the true surface of responses.

The beverages took scores in the range of 2.8 to 3.6 in a five-point hedonic scale. WP negatively affected OA at linear (b1 < 0.001) and quadratic levels (b11 < 0, *p* <0.05) (Table 3), whereas no significant effect of OF was observed at the linear level. Previous studies showed that an adequate thermal treatment is necessary for the development of oat aroma [60,61]. According to the consumer rating, high WP contents led to the detection of an unpleasant odor in fermented KLB samples and a decrease of OA. This result is in accordance with previous studies, where WP was associated to low sensory acceptance [30,62]. OA was also negatively affected at the quadratic level of OF (b22 < 0, *p* < 0.001) and at the interactive levels (WP/OF) (Figure 4). 

### 3.6. Verification of the Model

The optimum point was verified by using WP at 11.51% and OF at 4.77% for the validation of the model. The confirmation was repeated three times. Results are shown in Table 5. 

### 3.7. Effect of Fermentation on Bioactive and Nutritional Compounds of KLB

Eleven phenolics were identified and quantified into the unfermented optimized mixture, including phenolic acids and flavonoids (Figure 5). Gallic acid was the most represented compound (17.93 ± 0.33 mg L^−1^). 

After fermentation, the phenolics profile in KLB that resulted was modified with respect to the unfermented mixture (Figure 5). Main changes were related to the release of catechin and hyperoside, and to the conversion of phenolic acids to microbial derivatives, like dihydroferulic acid, ethyl catechol, and *p*-Vinyl phenol, respectively (Figure 5). The release of catechin and hyperoside was likely due to the degradation of phenolics-associated carbohydrates and proteins and to the breakdown of oligomeric flavonoids [31,63,64]. The metabolism of phenolic acids has been described in both lactic acid bacteria and yeasts [58,65,66,67]. Dihydroferulic acid is the reduced derivative of ferulic acid. The decarboxylation of p-coumaric acid led to *p*-Vinyl *p*-Vinyl phenol. Ethyl catechol resulted from the reduction of vinyl derivative of caffeic acid [65]. Most of these microbial derivatives exert biological activities [58], and they may contribute to the aroma attributes of fermented foods. *p*-Vinyl phenol is considered a food additive and is approved as a flavoring agent. Ethyl derivatives are considered the most important flavor components of fermented soy sauce, although they may cause off flavor in wine [68]. 

After fermentation, KLB resulted in higher (*p* < 0.05) protein digestibility (94% ± 1%) with respect to the unfermented mixture (89% ± 1%). Recent developments in functional beverages focused on protein intake, since cereals and others plant matrices contain several compounds that can prevent protein digestibility [68,69]. Proteolysis operated by the endogenous enzymes as well by microbial enzymatic activities likely affected the increased protein digestibility; however, the degradation of antinutritional factors, some of which are responsible for binding proteins, must be considered as co-responsible for such increase in fermented KLB.

## 4. Conclusions

Our results showed that central composite designs might represent a useful tool to optimize the formulation of a functional carob-based KLB fortified with low cost and highly nutritional matrices, like WP and OF. Such substrates modulated the microbial growth in a dose-depending manner, as well as the antiradical activity, TPC, and acceptability of fermented KLB. By using 11.51% WP and 4.77% OF, we obtained the KLB with the best fit of desirability. The optimized formulation resulted in KLB enriched in bioavailable phenolics derivatives and highly digestible proteins.

## Figures and Tables

**Figure 1 foods-10-00294-f001:**
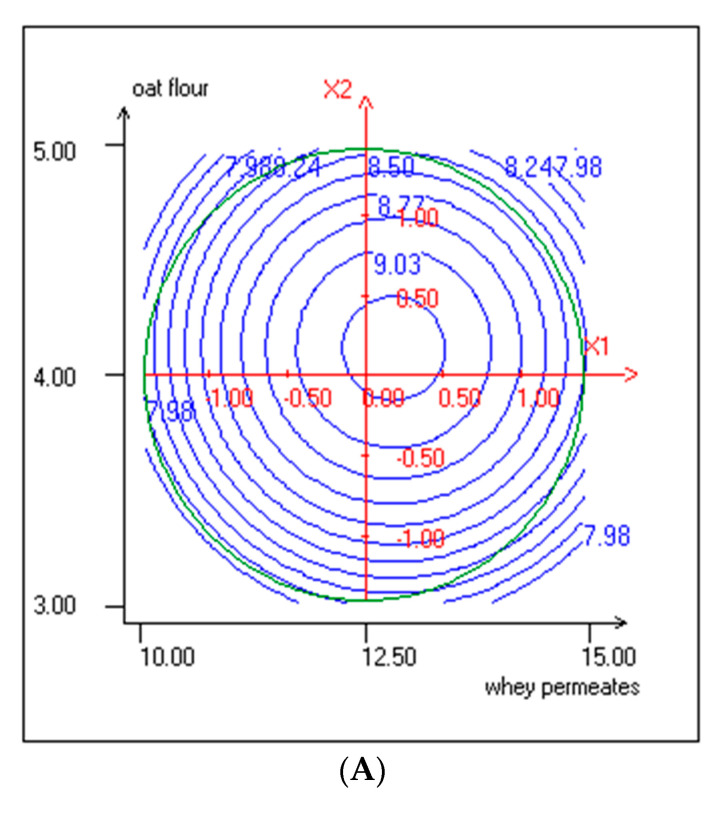
(**A**) Response surface and contour plots representing the effect of WP and OF on lactic acid bacteria cell counts (X1: WP; X2: OF). (**B**) Response surface and contour plots representing the effect of WP and OF on yeast cell counts (X1: WP; X2: OF).

**Figure 2 foods-10-00294-f002:**
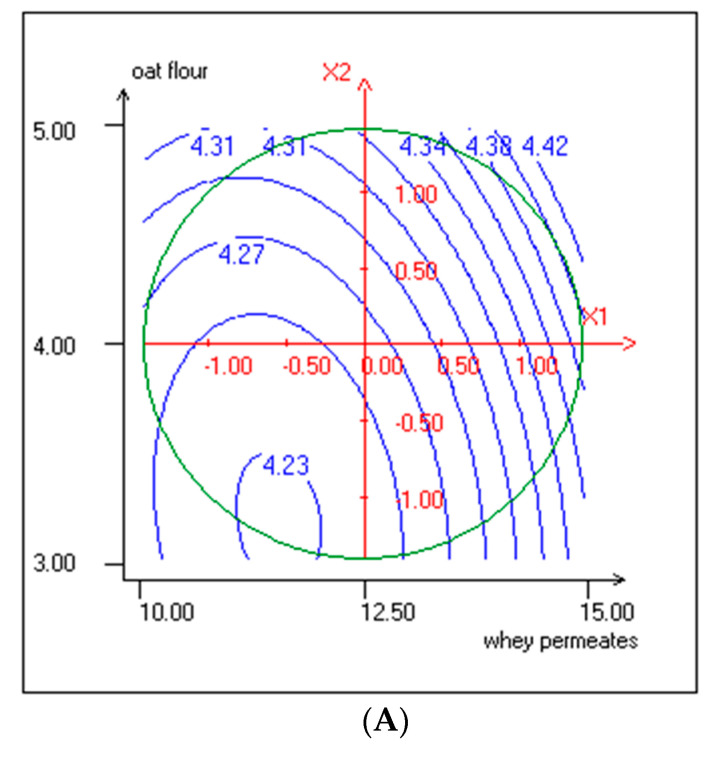
(**A**) Response surface and contour plots representing the effect of WP and OF on pH values (X1: WP; X2: OF). (**B**) Response surface and contour plots representing the effect of WP and OF on TTA (X1: WP; X2: OF).

**Figure 3 foods-10-00294-f003:**
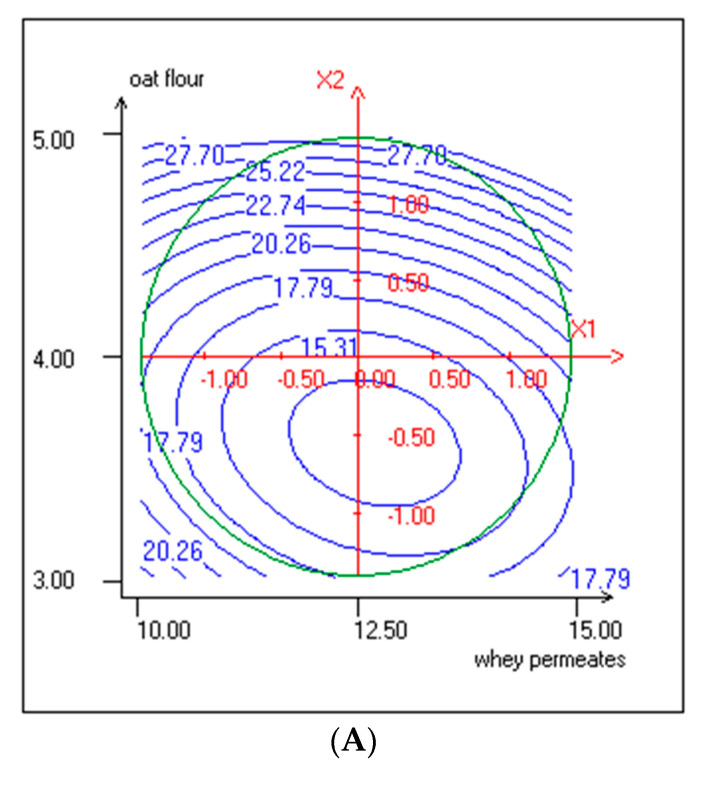
(**A**) Response surface and contour plots representing the effect of WP and OF on total polyphenol content (X1: WP; X2: OF). (**B**) Response surface and contour plots representing the effect of WP and OF on DPPH radical scavenging activity (%) (X1: WP; X2: OF).

**Figure 4 foods-10-00294-f004:**
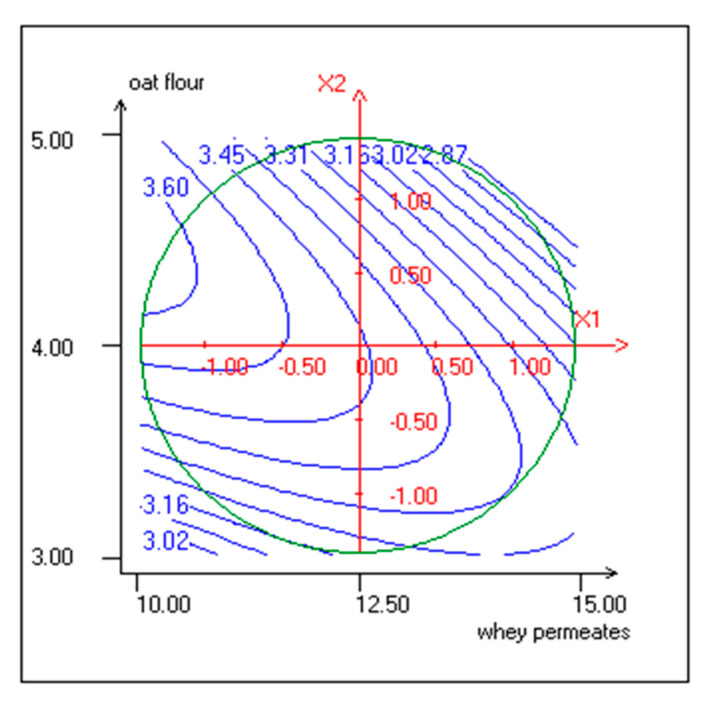
Response surface and contour plots representing the effect of WP and OF on overall acceptability (X1: WP; X2: OF).

**Figure 5 foods-10-00294-f005:**
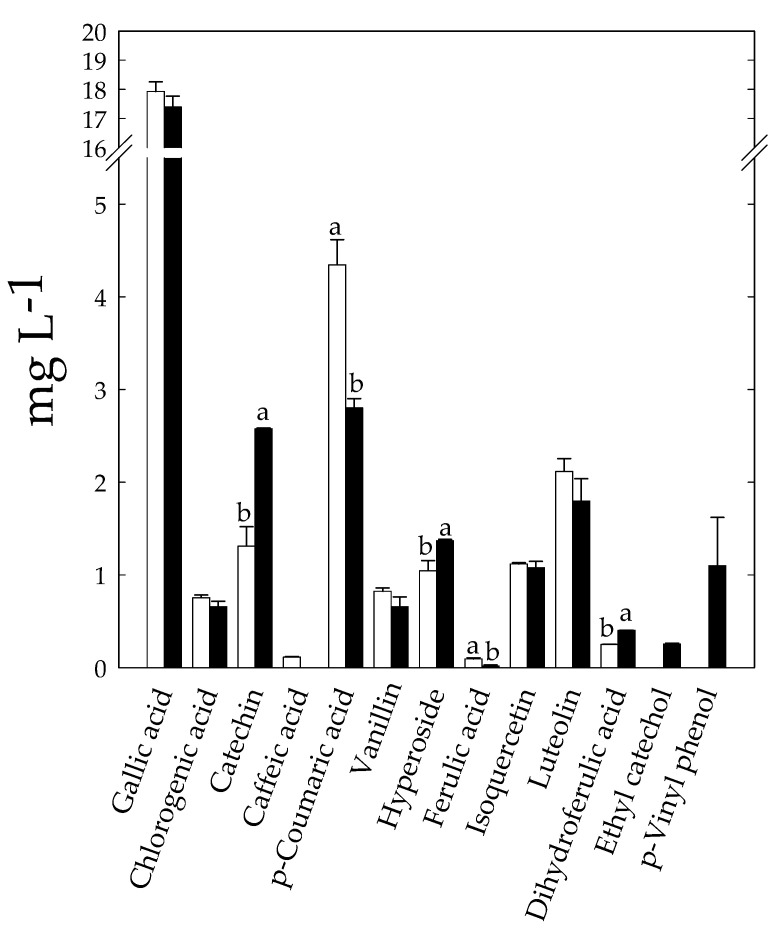
Concentration (mg L^−1^) of phenolic compounds identified into the unfermented mixture (white bars) and fermented KLB (black bars). Data were subjected to one-way ANOVA; pair-comparison of treatment means was achieved by Tukey’s procedure at *p* < 0.05. For each compound, bars with different superscript letters differ significantly (*p* < 0.05).

**Table 1 foods-10-00294-t001:** Independent variables (whey permeates, WP; oat flour, OF) and their levels in the central composite design.

Variable	Parameter (%, *w*/*v*)	Ranges and Level				
		−1.41	−1	0	1	1.41
X1	WP	10	10.73	12.5	14.27	15
X2	OF	3	3.29	4	4.71	5

**Table 2 foods-10-00294-t002:** Matrix of the central composite design (CCD) and observed responses to the percentages of WP and OF for the response variables (lactic acid bacteria (LAB) and yeasts cells count, pH, total titratable acidity (TTA), total phenolics content (TPC), DPPH radical scavenging activity, and overall acceptability (OA)).

ExpN°	Coded Variables				Responses Variables						
	WP (% *w*/*v*) Actual Values	Coded Values	Oat Flour (% *w*/*v*) Actual values	Coded Values	LAB Cells Count (log CFU/mL)	Yeasts Cells Count (log CFU/mL)	pH	TTA (%)	TPC (mg EGA/mL)	DPPH Radical Scavenging Activity (%)	OA (5 Point Hedonic Scale)
	X1		X2		(Y1)	(Y2)	(Y3)	(Y4)	(Y5)	(Y6)	
1	10.73 (−1)	−1	3.29	−1	7.85 ± 0.08	5.47 ± 0.27	4.21 ± 0.01	0.79 ± 0.02	19.365 ± 0.24	74.27 ± 0.33	3.20 ± 0.52
2	14.27 (1)	1	3.29	−1	8.17 ± 0.1	5.73 ± 0.24	4.29 ± 0.02	0.96 ± 0.02	17.369 ± 0.53	67.90 ± 0.023	3.30 ± 0.57
3	10.73 (−1)	−1	4.71	1	8.25 ± 011	5.70 ± 0.28	4.31 ± 0.03	0.86 ± 0.02	25.317 ± 0.28	81.84 ± 0.51	3.50 ± 0.61
4	14.27 (1)	1	4.71	1	8.55 ± 0.09	5.79 ± 0.26	4.42 ± 0.02	0.90 ± 0.02	27.697 ± 0.98	85.70 ± 0.39	2.80 ± 0.62
5	10.00	−1.41	4.00	0	8.02 ± 0.12	5.51 ± 0.29	4.26 ± 0.03	0.91 ± 0.04	15.625 ± 0.33	78.36 ± 0.29	3.60 ± 0.68
6	15.00	1.41	4.00	0	8.55 ± 0.1	6.00 ± 0.25	4.39 ± 0.02	0.92 ± 0.02	16.328 ± 0.3	75.30 ± 0.62	3.10 ± 0.52
7	12.50	0	3.00	−1.41	8.09 ± 0.09	5.69 ± 0.27	4.27 ± 0.04	0.90 ± 0.05	14.259 ± 0.38	71.65 ± 0.42	3.20 ± 0.41
8	12.50	0	5.00	1.41	8.45 ± 0.12	5.94 ± 0.29	4.29 ± 0.03	0.92 ± 0.02	24.617 ± 0.28	77.28 ± 0.25	3.10 ± 0.64
9	12.50	0	4.00	0	9.10 ± 0.10	6.05 ± 0.26	4.25 ± 0.02	1.05 ± 0.03	14.070 ± 0.41	74.61 ± 0.34	3.40 ± 0.50
10	12.50	0	4.00	0	9.11 ± 0.09	6.05 ± 0.29	4.27 ± 0.03	1.05 ± 0.05	14.780 ± 0.65	72.44 ± 0.29	3.45 ± 0.60
11	12.50	0	4.00	0	9.21 ± 0.10	6.04 ± 0.25	4.25 ± 0.02	1.05 ± 0.02	14.680 ± 0.52	73.38 ± 0.35	3.42 ± 0.53
12	12.50	0	4.00	0	9.19 ± 0.11	6.05 ± 0.28	4.27 ± 0.03	1.05 ± 0.03	14.680 ± 0.44	72.93 ± 0.22	3.55 ± 0.60
13	12.50	0	4.00	0	9.29 ± 0.09	6.05 ± 0.24	4.25 ± 0.01	1.07 ± 0.04	14.570 ± 0.32	72.69 ± 0.41	3.50 ± 0.61

TPC, Total polyphenol content mg EGA/mL. DPPH%, DPPH radical scavenging activity%.

**Table 3 foods-10-00294-t003:** Analysis of variance for each response Y1 (lactic acid bacteria cells count), Y2 (yeasts cell count), Y3 (pH), Y4 (TTA), Y5 (TPC), Y6 (DPPH radical scavenging activity), and Y7 (OA) (linear, quadratic and interaction).

SourceTitle	Y1		Y2		Y3		Y4		Y5		Y6		Y7	
	Coefficient	Signif.%	Coefficient	Signif.%	Coefficient	Signif.%	Coefficient	Signif.%	Coefficient	Signif.%	Coefficient	Signif.%	Coefficient	Signifi.%
Model														
b0	9.180	***	6.051	***	4.258	***	1.054	***	14.556	***	73.211	***	3.464	***
Linear														
b1	0.171	***	0.131	***	0.047	***	0.029	**	0.172	15.8	−0.855	*	−0.163	***
b2	0.161	***	0.080	***	0.032	**	0.006	14.7	3.866	***	4.167	***	−0.043	6.6
Quadratic														
b11	−0.466	***	-0.175	***	0.035	**	−0.078	***	1.893	***	2.255	**	−0.070	*
b22	−0.473	***	-0.146	***	0.012	*	−0.080	***	3.624	***	1.072	*	−0.170	***
Interaction														
b12	−0.005	90.1	-0.043	***	0.008	24.2	−0.032	**	1.094	**	2.557	**	−0.200	***
*R^2^*	0.986		0.925		0.861		0.920		0.835		0.816		0.965	
Adj-*R^2^*	0.975		0.872		0.762		0.862		0.717		0.684		0.940	

(* *p* < 0.05, ** *p* < 0.01, and *** *p* < 0.001).

**Table 4 foods-10-00294-t004:** DPPH radical scavenging activity and total phenolics content measured before and after KLB fermentation.

Independant Variables	DPPH Radical Scavenging Activity (%)		TPC (mg GAE/mL)	
	Unfermented	Fermented	Unfermented	Fermented
Exp 1	68.49 ^ef^ ± 0.67	74.27 ^de^ ± 0.46	9.59 ^a^ ± 0.34	19.36 ^e^ ± 0.55
Exp 2	65.60 ^a^ ± 0.44	67.90 ^a^ ± 0.55	14.04 ^f^ ± 0.44	17.36 ^d^ ± 0.37
Exp 3	70.53 ^g^ ± 0.61	81.84 ^g^ ± 0.89	10.73 ^b^ ± 0.52	25.31 ^f^ ± 0.63
Exp 4	68.84 ^ef^ ± 0.55	85.70 ^h^ ± 0.77	12.55 ^e^ ± 0.66	27.69 ^g^ ± 0.34
Exp 5	75.12 ^h^ ± 0.41	78.36 ^f^ ± 0.69	14.33 ^f^ ± 0.48	15.62 ^bc^ ± 0.62
Exp 6	68.43 ^de^ ± 0.52	75.30 ^e^ ± 0.55	11.75 ^cde^ ± 0.33	16.32 ^c^ ± 0.87
Exp 7	66.41 ^ab^ ± 0.52	71.65 ^b^ ± 0.73	10.88 ^bc^ ± 0.42	14.25 ^a^ ± 0.60
Exp 8	69.40 ^f^ ± 0.65	77.28 ^f^ ± 0.62	9.22 ^a^ ± 0.49	24.61 ^f^ ± 0.68
Exp 9	66.91 ^bc^ ± 0.43	74.61 ^e^ ± 0.51	11.50 ^bcd^ ± 0.68	14.07 ^a^ ± 0.34
Exp 10	66.86 ^bc^ ± 0.35	72.44 ^bc^ ± 0.87	11.60 ^bcd^ ± 0.55	14.78 ^ab^ ± 0.67
Exp 11	67.57 ^cd^ ± 0.63	73.38 ^cd^ ± 0.69	11.92 ^de^ ± 0.46	14.68 ^ab^ ± 0.47
Exp 12	67.17 ^bc^ ± 0.73	72.93 ^c^ ± 0.49	11.88 ^de^ ± 0.77	14.68 ^ab^ ± 0.72
Exp13	66.81 ^bc^ ± 0.45	72.69 ^bc^ ± 0.67	11.64 ^cd^ ± 0.62	14.57 ^a^ ± 0.56

Means ± standard error with the same letter did not differ significantly (*p* < 0.05). (Thirteen experiments as conducted by the central composite design representing thirteen KLB).

**Table 5 foods-10-00294-t005:** Predicted and experimental value for optimized formulation of KLB.

Response Variable	Optimum Conditions	
	Experimental	Predicted
pH	4.32 ± 0.02	4.29
TTA (%)	0.95 ± 0.04	0.96
Lactic acid bacteria cells count (Log CFU/mL)	8.62 ± 0.15	8.57
Yeasts cells count (Log CFU/mL)	5.80 ± 0.26	5.87
DPPH radical scavenging activity (%)	79.80 ± 0.46	78.59
TPC mg GAE/mL	21.40 ± 0.53	22.76
OA (5 point hedonic scale)	3.44 ± 0.46	3.41

Mean values ± SD (*n* = 3).

## Data Availability

The datasets generated for this study are available on request to the corresponding author.

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
