# Peer review of "Functional Exploitation of Carob, Oat Flour, and Whey Permeate as Substrates for a Novel Kefir-Like Fermented Beverage: An Optimized Formulation"

_foods, 2021, doi:10.3390/foods10020294_

Round 1
Reviewer 1 Report
Dear Authors,
Please find an enclosed review of the manuscript (Ms. Ref. No. foods-1067316), submitted by Sana M'hir, entitled “Functional exploitation of carob, oat flour, and whey permeate as substrates for a novel kefir-like fermented beverage: an optimized formulation”. Thank you for having me as a referee for the above-mentioned manuscript.
After studying the article given to me for evaluation, I state the following:
In the beginning, I would like to express words of my appreciation for the idea and effort put into conducting research and writing the manuscript recommended to me for review. The authors of the manuscript present a new study for the fortification of a carob-based kefir-like beverage with whey permeate and oat flour. In the manuscript, they show the effect of whey permeate and oat flour concentrations on lactic acid bacteria and yeast cell densities, pH, total titratable acidity, total phenolics content, DPPH radical scavenging activity, and overall acceptability. Authors also provide statistical designs for their findings.
However, after reading the manuscript, I have concerns regarding the issues listed below. I would like to ask you to elucidate the vagueness that emerged while reviewing the manuscript so you would be able to improve your investigation.
- A lot of statistical data presented in the manuscript without being confirmed in the form of experimental data gives the impression that the manuscript is a statistical document, rather than as the authors claim, an experimental work that compares and confirms their findings in the form of a statistical completion. Therefore, please add to the manuscript or supplement information related to the actual results of the experiments in the forms of graphs not only Tables.
- The average content of polyphenols is determined by means of a calibration curve determined for gallic acid at wavelengths of 725 nm and 760 nm according to the literature findings. In section 2.6, please explain why the 750 nm wavelength was used in the experiments, referring to other literary discoveries in this area in addition to the one given in this section (authors self-citation). Will the results correlate with each other if a different wavelength presented by the reviewer above is applied?
- DPPH is a well soluble chemical compound in organic solvents and insoluble in water. How was your manuscript measurements handled?
- In section 2.9, please give the chromatograms from the polyphenol analysis and complete this section with the conditions that were applied during the analysis on MS. How the data obtained from LCMS were useful for calculations in the manuscript and why LCMS analyses were performed when no phenolic compounds were identified and presented in the table with retention time and mass to charge ratio.
These are my main substantive concerns. Authors of the manuscript, please refer to them by showing additional results and discussion.
Good luck!
Author Response
- Point 1: A lot of statistical data presented in the manuscript without being confirmed in the form of experimental data gives the impression that the manuscript is a statistical document, rather than as the authors claim, an experimental work that compares and confirms their findings in the form of a statistical completion. Therefore, please add to the manuscript or supplement information related to the actual results of the experiments in the forms of graphs not only Tables.
Response 1: To facilitate the understanding of the results, a syntheses of the results obtained was carried out and the number of tables was reduced.
- The average content of polyphenols is determined by means of a calibration curve determined for gallic acid at wavelengths of 725 nm and 760 nm according to the literature findings. In section 2.6, please explain why the 750 nm wavelength was used in the experiments, referring to other literary discoveries in this area in addition to the one given in this section (authors self-citation). Will the results correlate with each other if a different wavelength presented by the reviewer above is applied?
Response 2: Phenolic compounds’ analysis was carried out using the technique described in the article of Karaaslan et al. 2011: the absorbance was measured at 750 nm. Normally, there will not be a significant difference between the results obtained at 750nm and 760 nm.
- Karaaslan, M. Ozden, H. Vardin, and H. Turkoglu, “Phenolic fortification of yogurt using grape and callus extracts,” LWT—Food Science and Technology, vol. 44, no. 4, pp. 1065–1072, 2011. https://doi.org/10.1016/j.lwt.2010.12.009
- DPPH is a well soluble chemical compound in organic solvents and insoluble in water. How was your manuscript measurements handled?
Response 3: The extraction of phenolic compounds contained in the juices studied was carried out by the Methanol-water mixture: 80-20 (v/v). Then, the anti-radical activity of the various extracted compounds was evaluated in vitro, by the test at the DPPH.
- In section 2.9, please give the chromatograms from the polyphenol analysis and complete this section with the conditions that were applied during the analysis on MS. How the data obtained from LCMS were useful for calculations in the manuscript and why LCMS analyses were performed when no phenolic compounds were identified and presented in the table with retention time and mass to charge ratio.
Response 4:
Ok, MS method details were added (P4 L168-176). A table showing the retention time and m/z for compounds identified by LCMS and a figure showing the UV chromatograms were provided as supplementary files (please, see new Table S1 and new Figure S1). We preferred not to show them into the main manuscript, due to the high number of figures and tables already reported. However, the Figure 5 within the main manuscript shows the quantification of each phenolic compound. LCMS data were useful to identify and quantify phenolics compounds in KLB (Figure 5), as they may exert potential health-promoting effects (P13 L134-161).

Reviewer 2 Report
The study presented by authors aimed the optimization of the KLB formulation using the response surface methodology. The manuscript is clear and well written. I suggest to check the following minors to improve it.
- Some small typos (materials and methods 2.2 an space between the number and the %, and between 90 ºC, and 30 ºC. Check through the whole manuscript, there are several.
- It is enough to declare abbreviations once. WP, OF, TTA and TCP are declared several times. However, there are some that are not declare: eg. PDA, MRS.
- Use the same rule for in vivo and in vitro.
- Tittles in the table 3, some are in bold and some not.
- Avoid to use the term probiotic when referring to a microbial composition of the kefir grains. It is better to use health-promoting bacteria than probiotic.
- It would be really interesting to characterize the microbial composition of the kefir grains.
- Since there is not a separate discussion from the results, I missed some comparative analysis with previous studies.
Author Response
Point 1 : Some small typos (materials and methods 2.2 à an space between the number and the %, and between 90 ºC, and 30 ºC. Check through the whole manuscript, there are several.
Response 1 : The spaces were checked
Point 2 : It is enough to declare abbreviations once. WP, OF, TTA and TCP are declared several times. However, there are some that are not declare: eg. PDA, MRS.
Response 2 : The abbreviations for medium were checked. « PDA » and « MRS » were declared :
PDA : Potate Dextrose Agar
MRS : De Man, Rogosa and Sharpe.
- Point 3 : Use the same rule for in vivo and in vitro.
Response 3 : Ok
- Point 4 : Tittles in the table 3, some are in bold and some not.
Response 4 : The titles were checked
- Point 5 :Avoid to use the term probiotic when referring to a microbial composition of the kefir grains. It is better to use health-promoting bacteria than probiotic.
Response 5: The sentence was corrceted
« For instance, kefir grains were used to ferment non-dairy substrates, such as tea and plant juices, to produce functional beverages enriched in bioactive compounds or able to vehicle potential health-promoting bacteria »
- Point 6: It would be really interesting to characterize the microbial composition of the kefir grains.
Response 6 : we have tried preliminary composition.
Enumeration of LAB was done on M17-lactose and MRS agars (Merck, Germany) to estimate the numbers of presumptive coccus and bacillus, respectively. Preliminary identification of LAB was on the basis on cell morphology and phenotypic properties Pure isolates of LAB and yeasts were inoculated into API 50 CHL and API 20 AUX kits respectively, to study carbohydrate metabolism profiles (Biomerieux, France).The starter culture consists of a symbiotic consortium of several yeasts and bacteria. LAB are represented by the genera of Leuconostoc, Lactobacillus, and Lactococcus.. About 46% with bacilli form, 27% coccobacille form, and 27 % cocci. Yeasts include the genera of Saccharomyces and Zygosaccharomyces were identified, according to the criteria of Sievers et al. (1995).
- Point 7 : Since there is not a separate discussion from the results, I missed some comparative analysis with previous studies.
Response 7 : The discussion was improved
Reviewer 3 Report
The paper aimed at evaluating the applicability of central composite design as a tool for the optimization of KLB formulation in order to obtained a fortified functional formulation. The paper is well designed and structured as well as well written and easy to understand.
Only few comments:
- Pag 2: Under the conditions of our study.. I suggest to modified in “in our study”
- At pag 2 : Fortification of functional beverages with prebiotic such as OF [17] could improve their nutritional proprieties as well to… please correct the font
- I also suggest to rewrite the aim of the paper in a more understandable way and move it at the end of the introduction. I think that it should be better explained that a model was used to choose the variable tested for the fermentation.
- Pag 2: Materiel and Methods please correct
- Kefir grains belonged to LETMi laboratory could you please provide more information about the grains? Do you know which yeast and bacterial species are presents? I think that this could be a very interesting information
- About the ingredients: Do you know the composition of WP? How much sugar is present? Because at pag. 8 you state that lactose is the carbon source for the fermentation. So I think that this could be a very important information. The same for the OF. Is evident that if no sugar is added the success of fermentation is due only to the sugar presents in the added ingredients, therefore if the percentage used are different, as a consequence also the amount of available sugars may vary, influencing the fermentation. For this reason I suggest to add as much information as possible about the ingredients used.
- Do you know which was the balance between yeast and bacteria after fermentation? Because the different ingredients, that for sure may provide different substrates for the fermentation, could lead to a different balance between yeast and m.o. and thus to differences in the final product.
- Did you performed a microbiological analysis before and after fermentation? This question because you added a non-sterile ingredients that in some way may contribute to the fermentation. I know that was not the aim of the study but, could you provide this information?
Author Response
- Point1: Pag 2: Under the conditions of our study.. I suggest to modified in “in our study”
Response 1:Under the conditions of our study was modified by I “in our study”
- Point 2: At pag 2 : Fortification of functional beverages with prebiotic such as OF [17] could improve their nutritional proprieties as well to…please correct the font
Response 2: the italics form was deleted
- Point 3: I also suggest to rewrite the aim of the paper in a more understandable way and move it at the end of the introduction. I think that it should be better explained that a model was used to choose the variable tested for the fermentation.
Response 3: The aim of this study was rewritten and placed at the end of the introduction
The aim of this study was to optimize the formulation of a functional KLB, which was fortified with various amount of WP and OF to improve the health-promoting features and sensory properties of KLB. A central composite design (CCD) was used to analyze the interactions between the independent variables, producing a mathematical model describing the investigated system. This approach allowed to determine the optimal WP and OF supplementation levels to maximize the cell density of lactic acid bacteria and yeasts, the DPPH radical scavenging activities, the total phenolics content, and the consumer acceptability of KLB. Phenolic profile and protein digestibility were further evaluated for the optimized beverage.
- Point 4: Pag 2: Materiel and Methods please correct
Response 4 : The title has been revised
- Point 5: Kefir grains belonged to LETMi laboratory could you please provide more information about the grains? Do you know which yeast and bacterial species are presents? I think that this could be a very interesting information
Response 5: As previously characterized, kefir grains consisted of a symbiotic consortium of lactic acid bacteria (Leuconostoc spp., Lactobacillus spp., and Lactococcus spp.) and yeasts (Saccharomyces spp. and Zygosaccharomyces spp.,).
- Point 6: About the ingredients: Do you know the composition of WP? How much sugar is present? Because at pag. 8 you state that lactose is the carbon source for the fermentation. So I think that this could be a very important information. The same for the OF. Is evident that if no sugar is added the success of fermentation is due only to the sugar presents in the added ingredients, therefore if the percentage used are different, as a consequence also the amount of available sugars may vary, influencing the fermentation. For this reason I suggest to add as much information as possible about the ingredients used.
Response 6: The following composition of raw material was provided within the manuscript:
Whey permeates were supplied from a cheese making company. It had the following characteristics: proteins 3%, lactose 85%, and ash 7%.
Oat flour : 14.8 % proteins, carbohydrates 66.3 %
Carob decoction: 15 °Brix
- Point 7: Do you know which was the balance between yeast and bacteria after fermentation? Because the different ingredients, that for sure may provide different substrates for the fermentation, could lead to a different balance between yeast and m.o. and thus to differences in the final product.
Response 7: As reported by Laureys and De Vuyst (2014), there were 2 to 10 LAB for each yeast cell on water kefir grains. However, the ratio changed during fermentation depending on the concentration of WP and OF used: It was, for example, 239 (experiment 1), 275 (experiment 2), and 354 (experiment 6). The experiments were chosen randomly for each study.
- Point 8: Did you performed a microbiological analysis before and after fermentation? This question because you added a non-sterile ingredients that in some way may contribute to the fermentation. I know that was not the aim of the study but, could you provide this information?
Response 8: All beverages were pasteurized before the addition of kefir grains.

Reviewer 4 Report
This research investigated the fortification of a carob-based kefir-like beverage with whey permeate and oat flour. There are numerous confusions in the statistical annotation. For example, Ecart-type should be Standard Errors, Signif. % should be Significance (% value should be changed the value), t.exp -> T value,
In Table 2, X1 and X2 values assigned to each run are uncoded value. For a better understanding of experimental design, coded values (-1, 0, 1) should be provided.
Most of the tables are not fit journal format (alignment, decimal, lack of appropriate foot not, unusual abbreviation).
In Table 4, there is significance in lack of fit. Authors should change model analysis under consideration of lack of fit. or should mention the consideration of lack of fit.
In figures, serious revisions are required. There are two different plots (RSM and contour) with the same meaning. In the contour plot, two values (coded and un-coded values of independent variables) are annotated together. And values with blue color annotated on the contour plot is not easy to understand. Visualization is one of the most important ways and easy to figure out the result. However, Figure 1 is very complicated and not easy to read the values.
Table 5, independent variables are Exp. No in Table 2. And the author already tested the hypothesis in Table 4. But, the author conducted multiple comparisons between runs (O1 ~ O13). And there is no footnote explaining superscript(same row or column ?).
There are serious problems in statistical analysis, understanding, and notation.
Author Response
Point 1: This research investigated the fortification of a carob-based kefir-like beverage with whey permeate and oat flour. There are numerous confusions in the statistical annotation. For example, Ecart-type should be Standard Errors, Signif. % should be Significance (% value should be changed the value), t.exp -> T value,
Response 1: The statistical annotation was improved.Point 2: In Table 2, X1 and X2 values assigned to each run are uncoded value. For a better understanding of experimental design, coded values (-1, 0, 1) should be provided. Most of the tables are not fit journal format (alignment, decimal, lack of appropriate foot not, unusual abbreviation).
Response 2: The tables were improved, and the coded values were added.Point 3: In Table 4, there is significance in lack of fit. Authors should change model analysis under consideration of lack of fit. or should mention the consideration of lack of fit.
Response 3: Analysis of variance (ANOVA) of the quadratic regression model shows that the models are highly significant (P< 0.01) for lactic acid bacteria growth and P< 0.05 for overall acceptability. This is justified by the fact that the value of the calculated F is much greater than the tabled F value.
Probability (P-values) indicates the meaning of the model. The P-value relative to the regression model is very low, which shows the meaning of the model. In addition, the contribution of each of the linear, quadratic and interaction terms is significant at the 5% threshold. Therefore, it can be concluded that the prediction models are highly significant and adequately estimate the data observed.
Furthermore, there is a significant repeatability compared to the central points. R square and adj-R square had values greater than 0.95; and the contours' plots show quite well defined optimal areas.
Point 4 : In figures, serious revisions are required. There are two different plots (RSM and contour) with the same meaning. In the contour plot, two values (coded and un-coded values of independent variables) are annotated together. And values with blue color annotated on the contour plot is not easy to understand. Visualization is one of the most important ways and easy to figure out the result. However, Figure 1 is very complicated and not easy to read the values.
Response 4 : Only figures with plots have been maintained. These figures were given by the software (Nemrod), and we cannot do any modifications.
Point 5: Table 5, independent variables are Exp. No in Table 2. And the author already tested the hypothesis in Table 4. But, the author conducted multiple comparisons between runs (O1 ~ O13). And there is no footnote explaining superscript (same row or column ?). There are serious problems in statistical analysis, understanding, and notation.
Response 5: Tables have been modified to be clearer and additional explanations have been added to facilitate understanding.

Reviewer 5 Report
The submitted work is dealing with an interesting topic regarding an “optimization” attempt to formulate a functional fermented beverage. Even though the study is designed appropriately and most observations are adequately supported, when needed, by the discussion (comparing results with other scientific data) some points and results are not clearly presented. Hence, some more specific comments and recommendations are provided below:
The authors used Central Composite Design for this work, thus a respective reference should be added (in section 2.3), while equations of second-degree polynomial model used, should be numbered and presented in the “Materials & Methods” section (instead of the Results section 3.2, 3.3, 3.4).
A first critical remark concerns the use of whey permeate (defined as “WP”, in the manuscript). Authors state ‘’WP, a by-product of the cheese-making industry, is rich in peptides and minerals. Adding WP to foods or beverage improves their amino acid profile because more than 50% of its amino acids are essentials’’ plus ‘’WP was used due to its high content of proteins and oligosaccharides’’. However, it is widely known that whey permeate is a high lactose fraction in contrast to whey protein fraction which is rich in proteins, peptides and essential amino acids. The authors should reconsider this issue throughout their manuscript. Moreover, the authors state ‘’ Carob is high in sugar content but poor in fatty acids and proteins’’ plus ‘’Carob is an excellent source of phytochemicals including proteins and amino acids, fatty acids, carbohydrates and polyphenolic compounds’’. The authors are recommended to properly revise these contradictory statements in their manuscript. In addition, the following part should be rewritten since it is misleading for readers, ‘’the enrichment of carob pods decoction with WP and OF showed a positive effect on biomass production. It was likely due to the low content of protein of carob, which was increased by the addition of WP or the OF (showed by preliminary experiment)’’.
My second critical remark concerns the experimental analysis. The authors used the Response Surface Methodology as a statistical approach in their study; therefore, the presentation of the observations using pure method terminology is in generalyl confusing for the reader to deal with a clear result. ‘’The pH and TTA were found to be in function of the linear and quadratic effects of WP and OF. The linear and quadratic effects were both positive for pH, which explained the observed behavior of the curve as shown in Figure 2A. The linear effects were positive, whereas the quadratic effects were negative for TTA, which resulted in a curvilinear increase in TTA for all the WP and OF concentrations’’. The authors are advised to revise such descriptions in a more “reader-firendly” way.
At last, the author should consider revising the Analysis of Variance Table 4. In specific, all five-response variables used could be merged into one single table in order to be in a clear and easily comparable format. Furthermore, means square, residual and pure error could be removed (and maybe provided as supplementary material) as they make results presentation overloaded. The independent variables should be also indicated, especially if they concern the interaction of WP and OF.
Author Response
Point 1: The authors used Central Composite Design for this work, thus a respective reference should be added (in section 2.3), while equations of second-degree polynomial model used, should be numbered and presented in the “Materials & Methods” section (instead of the Results section 3.2, 3.3, 3.4).
Response 1: The reference about RSM was added:
- Mayer, R.; Montgomery, D. Response Surface Methodology: Process and Product Optimization Using Designed Experiments; JohnWiley and Sons, Inc.: New York, NY, USA, 1995
The equations for LAB, yeast, pH, Total tirable acidity, TPC, DPPH scavenging activity and OA were reported in MM section instead of Results
Point 2: A first critical remark concerns the use of whey permeate (defined as “WP”, in the manuscript). Authors state ‘’WP, a by-product of the cheese-making industry, is rich in peptides and minerals. Adding WP to foods or beverage improves their amino acid profile because more than 50% of its amino acids are essentials’’ plus ‘’WP was used due to its high content of proteins and oligosaccharides’’. However, it is widely known that whey permeate is a high lactose fraction in contrast to whey protein fraction which is rich in proteins, peptides and essential amino acids. The authors should reconsider this issue throughout their manuscript.
Response 2: The sentences about WP were corrected. Whey permeates had the following characteristics: proteins 3%, lactose 85%, and ash 7%. That’s why the oat flour was used to fortify the beverage with protein.
The paragraph has been amended as follows:
« WP, a by-product of the cheese-making industry, was previously described as valuable growth medium for probiotic bacteria, able to ensure a high microbial viability [22]. The use of WP as natural antioxidant in foods was also proposed [23]. Furthermore, the addition of whey to fermented products allows the enhancement of prebiotic properties [24]. «
Moreover, the authors state ‘’ Carob is high in sugar content but poor in fatty acids and proteins’’ plus ‘’Carob is an excellent source of phytochemicals including proteins and amino acids, fatty acids, carbohydrates and polyphenolic compounds’’. The authors are recommended to properly revise these contradictory statements in their manuscript.
Response 3: The sentence was corrected to avoid contradiction
Point 4: In addition, the following part should be rewritten since it is misleading for readers, ‘’the enrichment of carob pods decoction with WP and OF showed a positive effect on biomass production. It was likely due to the low content of protein of carob, which was increased by the addition of WP or the OF (showed by preliminary experiment)’’.
Response 4: This paragraph has been reformulated to facilitate understanding.
Lactic acid bacteria and yeasts cell densities varied from 7.85 ± 0.08 to 9.29 ± 0.09 and 5.47 ±0.27 to 6.05 ± 0.29 log CFU/mL, respectively (Table 2). The enrichment of carob decoction with OF and WP has led to a significant improvement in growth. In fact, when carob was used as sole carbon and nitrogen source (data not shown), the lactic acid bacteria growth was limited. Bouhadj et al. [15] showed an increase of lactic acid bacteria growth and lactate production when the carob syrup was enriched with sweet cheese. The promoting effect of oat on the growth of probiotic bacteria and their use as a probiotic carrier has also been demonstrated in many studies [39-41].
Point 5: My second critical remark concerns the experimental analysis. The authors used the Response Surface Methodology as a statistical approach in their study; therefore, the presentation of the observations using pure method terminology is in generalyl confusing for the reader to deal with a clear result. ‘’The pH and TTA were found to be in function of the linear and quadratic effects of WP and OF. The linear and quadratic effects were both positive for pH, which explained the observed behavior of the curve as shown in Figure 2A. The linear effects were positive, whereas the quadratic effects were negative for TTA, which resulted in a curvilinear increase in TTA for all the WP and OF concentrations’’. The authors are advised to revise such descriptions in a more “reader-friendly” way.
Response 5: This paragraph has been reformulated to facilitate understanding.
The pH and TTA were found to be in function of the linear and quadratic effects of WP and OF. As shown by linear coefficients, WP and OF affected positively the pH and TTA. Nevertheless, the quadratic coefficients of WP and OF were negative for TTA. On the other hand, the interactive effect between WP and OF was not significant for pH but it was significant for TTA (Table 3).
Point 6: At last, the author should consider revising the Analysis of Variance Table 4. In specific, all five-response variables used could be merged into one single table in order to be in a clear and easily comparable format. Furthermore, means square, residual and pure error could be removed (and maybe provided as supplementary material) as they make results presentation overloaded. The independent variables should be also indicated, especially if they concern the interaction of WP and OF.
Response 6: Given the large amount of statistical analysis Table 4 was transferred to supplementary material: Table S2 and the R2 values and adj R2-adj were added to the Table 3

Round 2
Reviewer 1 Report
Accept
Reviewer 5 Report
The authors have followed reviewers' suggestions and significantly improved the manuscript.